# Unveiling the Inflammatory Landscape of Recurrent Glioblastoma through Histological-Based Assessments

**DOI:** 10.3390/cancers16193283

**Published:** 2024-09-26

**Authors:** Nicholas B. Dadario, Deborah M. Boyett, Damian E. Teasley, Peter J. Chabot, Nathan J. Winans, Michael G. Argenziano, Colin P. Sperring, Peter Canoll, Jeffrey N. Bruce

**Affiliations:** 1Department of Neurological Surgery, Columbia University Irving Medical Center, NY-Presbyterian Hospital, New York, NY 10032, USA; dmb2245@cumc.columbia.edu (D.M.B.); det2130@cumc.columbia.edu (D.E.T.); pjc2194@cumc.columbia.edu (P.J.C.); njw2130@cumc.columbia.edu (N.J.W.); mga2122@cumc.columbia.edu (M.G.A.); cps2155@cumc.columbia.edu (C.P.S.); pc561@cumc.columbia.edu (P.C.); 2Department of Pathology and Cell Biology, Columbia University Irving Medical Center, NY-Presbyterian Hospital, New York, NY 10032, USA

**Keywords:** recurrent glioblastoma, immune cells, tumor microenvironment, histology, microglia, macrophages

## Abstract

**Simple Summary:**

Glioblastoma is a highly aggressive tumor, and despite standard-of-care therapy, tumor recurrence is inevitable. Recurrent glioblastoma (rGBM) remains a challenge to treat, given that it predominantly forms within a highly inflammatory tumor microenvironment (TME) that contributes to treatment resistance. However, the complex immune landscape of rGBM and how to effectively characterize it for therapeutic targeting remains poorly understood. To address this gap, this review exhaustively examines the existing body of literature on the immune characteristics of rGBM with a focus on the role of glioma-associated microglia and macrophages (GAMMs). We examine this inflammatory landscape through the lens of histological-based assessments given histological staining remains the gold standard of diagnostic identification of rGBM. We review canonical and emerging cell-specific markers of GAMMs and provide a guide of how these histological-based assessments can aid in characterizing the TME of rGBM and dissect how this landscape shifts in the context of treatment response.

**Abstract:**

The glioblastoma (GBM) tumor microenvironment consists of a heterogeneous mixture of neoplastic and non-neoplastic cells, including immune cells. Tumor recurrence following standard-of-care therapy results in a rich landscape of inflammatory cells throughout the glioma-infiltrated cortex. Immune cells consisting of glioma-associated macrophages and microglia (GAMMs) overwhelmingly constitute the bulk of the recurrent glioblastoma (rGBM) microenvironment, in comparison to the highly cellular and proliferative tumor microenvironment characteristic of primary GBM. These immune cells dynamically interact within the tumor microenvironment and can contribute to disease progression and therapy resistance while also providing novel targets for emerging immunotherapies. Within these varying contexts, histological-based assessments of immune cells in rGBM, including immunohistochemistry (IHC) and immunofluorescence (IF), offer a critical way to visualize and examine the inflammatory landscape. Here, we exhaustively review the available body of literature on the inflammatory landscape in rGBM as identified through histological-based assessments. We highlight the heterogeneity of immune cells throughout the glioma-infiltrated cortex with a focus on microglia and macrophages, drawing insights from canonical and novel immune-cell histological markers to estimate cell phenotypes and function. Lastly, we discuss opportunities for immunomodulatory treatments aiming to harness the inflammatory landscape in rGBM.

## 1. Introduction

Glioblastoma (GBM) is the most common primary brain malignancy, with a median overall survival of 14.6 months following standard-of-care therapy [1]. Despite gross total resection of the contrast-enhancing tumor and adjuvant chemoradiotherapy, a residual mixture of invasive neoplastic and non-neoplastic cells is left, which forms the basis for inevitable recurrence [2]. Recurrent GBM (rGBM) demonstrates a unique tumor microenvironment (TME) that differs significantly from primary GBM [3]. Unlike primary GBM, rGBM is often not a highly cellular and proliferative tumor [3]. After various therapeutic pressures following standard-of-care therapy and competitive selection among varying cell populations, rGBM forms within a highly inflammatory microenvironment that favors therapeutic resistance [4]. In fact, the amount of immune infiltrate in rGBM often outnumbers the degree of tumor burden [5]. This robust inflammatory response can be observed on a macroscopic and radiographic level, commonly termed “treatment effect” [6]. Since most studies examining the inflammatory landscape of glioma have focused on primary GBM patients, a significant gap in knowledge about the immune microenvironment exists in rGBM patients.

The ability to harness the immune response has resulted in significant anti-tumor effects and prolonged clinical efficacy in various cancers [7]. Despite this success, immunotherapy has largely failed to be efficacious for GBM [8] in part due to the predominantly “immunosuppressive” tumor microenvironment, which can limit effective treatment [9]. However, many cytotoxic drugs are also known to cause a robust inflammatory microenvironment, which can also contribute to the failure of therapy, including in recurrent GBM [10]. Ultimately, immune cells demonstrate significant cross-talk within the glioma microenvironment, leading to glioma progression and treatment failure [11]. A better understanding of the immune microenvironment in glioma is needed for potential therapeutic benefit, especially in highly inflammatory rGBM. One common way to better understand the inflammatory landscape in human tissue is through histopathological characterization [12]. Through novel histological analyses often guided by RNA-sequencing-based signatures, various large case series have characterized inflammation in glioma-infiltrated tissue and tissue-based treatment responses to emerging therapies [3,5,13,14,15,16,17,18,19,20]. However, the inflammatory landscape in rGBM and the applicability of using histological-based assessments in these patients have yet to be reviewed in detail.

In this review, we discuss the available literature on histological-based assessments of rGBM tissue and review the current understanding of the immunoinflammatory landscape in rGBM. We focus specifically on how clinically available, canonical, and emerging cell-specific markers could aid in histological analyses in rGBM. We highlight the role of microglia and macrophages in this setting and our institution’s experience in examining these significant cell populations. These findings may provide a guide for histological-based assessments in the neuro-oncological community moving forward in this critical patient population.

## 2. The Immune Microenvironment: Key Players

Previous studies have examined paired primary-recurrent GBM samples to better understand the mechanisms of rGBM evolution with a focus on tumor cells [5]. However, they have varied in response to the supposed degree of overlap in clonal expansion, genomic drivers, and other molecular features, which may cause the primary tumor to evolve into a recurrent tumor after treatment [12,21]. Like primary GBM, rGBM also contains various tumor subtypes, which are often retained upon recurrence [5]. However, these tumor subtypes also have cross-talk with the immune microenvironment [11], and in turn, heterogeneity in immune profiles contributes to heterogeneity in tumor populations in rGBM as well [5,21]. Recurrent tumors, in particular, demonstrate a preferential mesenchymal progression [5], a phenotype that is highly inflammatory and associated with a less favorable clinical outcome compared to other tumor/immune subtypes [21]. Rather than a shift or evolution of tumor phenotype alone towards a mesenchymal subtype upon tumor recurrence, this induction may also be driven by changes in the tissue composition of the TME, such as with immune cells [2,5]. Microglia and macrophages in the TME are believed to largely influence the mesenchymal phenotype of a tumor, such as by the relative abundance of macrophages to tumor cells or by direct downstream signaling pathways [22]. This transition may be mediated through various macrophage-mediated processes, such as inactivation of NF1 and hypoxia, a reduction in tumor purity alongside macrophage expansion, or by macrophage-mediated changes in tissue composition, such as cytotoxicity towards T-cells [22].

Despite this complexity, what is clear is that rGBM is characterized by an increase in immune cell activation and infiltration in many cases upon disease recurrence. Therefore, it is critical to first understand what the immune cells are that make up the glioma microenvironment to better understand the rGBM TME.

The glioma tumor microenvironment (TME) predominantly consists of glioma cells, neurons, vascular cells, and glia. The immune component of the TME can largely be characterized by microglia, macrophages, leukocytes (CD4+ T helper cells, CD8+, Treg), natural killer cells, and myeloid-derived suppressor cells. While different immune or tumor subtypes are important and likely exist in varying degrees between groups of patients, tumor-associated macrophages and/or microglia constitute the majority of immune cells across most rGBM patients and tumor/immune subtypes (Figure 1). We focus on these cell types below, including common histological identifiers, differences in expression profiles, and their clinical significance.

## 3. Glioma-Associated Macrophages and Microglia (GAMM)

Glioma-associated macrophages and microglia (GAMM) make up the predominant immune cell type in recurrent glioma. Similarly, in primary GBM, GAMMs can constitute up to 30–40% of the TME [23]. Although no clear estimate exists in rGBM, GAMMs likely play a key role in recurrent glioma maintenance and progression [24].

### 3.1. Origins

Microglia are resident immune cells of the central nervous system (CNS) responsible for continuous immune surveillance throughout the CNS and are the first responders to brain pathogens or injury in the form of microgliosis. They can constitute up to 15% of the total cell population in the human brain [25]. Macrophages, on the other hand, can be found throughout the entire human body, including beyond CNS tissue. Both cell types are believed to play an essential role in phagocytosis to maintain tissue homeostasis.

The origin of microglia and/or macrophages has been a topic of significant debate. Traditionally, microglia were thought to originate from either mesodermal cells of the pia mater, which then invades the brain parenchyma, or peripheral circulating monocytes [26], largely because increased ramified microglia appeared alongside signs of brain vascularization [27]. However, subsequent studies suggested that all microglia originate in the yolk sac (YS) from erythro-myeloid progenitor cells, or YS-macrophages (see review in [28]), which then populate the brain parenchyma. Conversely, tissue-resident macrophages are believed to originate from a fetal pool of hemopoietic stem cells. This supposed dichotomy is contradicted by recent data that up to 25% of all CNS microglia arise from fetal liver-derived monocytes [29]. This population of microglia is thought to be “macrophage-like” microglia, expressing markers of both macrophages and microglia.

Taken together, these works suggest that macrophages and microglia may exist in part on a spectrum, with context-dependent changes causing differing histological and expression levels between these entities (Figure 1). As we will explore in greater detail below, we refer in this work to ‘microglia’ as CNS-derived glial cells that express markers associated with resting microglia, whereas ‘macrophages’ encompass both blood-derived macrophages and macrophage-like microglia, or microglia which have activated and may shift towards a macrophage-like state. Thus, characterizing the rGBM inflammatory landscape, how it changes after treatment, and its clinical prognostic significance will require a combination of multiple markers and multiple modalities of analysis to histologically characterize these cell populations.

### 3.2. Histological Identification and Presence in rGBM

Histological visualization of GAMMs in tumor-infiltrated tissue has improved our understanding of their presence in primary glioma and is necessary to characterize rGBM, to measure the response to standard-of-care treatments, and to assess emerging therapies. Histological-based analyses, including immunohistochemistry (IHC) and multiplex immunofluorescence (IF), can clarify the potential identity (e.g., microglia, macrophage, or macrophage-like microglia), activity (e.g., resting versus active), and current phenotypic or functional state (e.g., phagocytosing and proliferative) of these immune cells.

It is important to note that the histological markers discussed below have largely benefited from data provided by single-cell “omics” analyses on immune cell-specific signatures [30,31], such as in an attempt to differentiate microglia and macrophages (Figure 1). For instance, single-cell and bulk sequencing analyses have commonly suggested microglia express core markers that differ from infiltrating macrophages, such as Cx3cr1, Hexb, Sall1, P2ry12, and Tmem119 [7,8,32]. However, histological-based assessments are necessary to validate these RNA measurements on the protein level (Figure 2) and can further supplement their discovery in a faster and often more cost-efficient manner as well as provide meaningful insight into spatially distinct subpopulations revealed in the tissue architecture [33]. Histological staining also remains the gold standard of diagnostic identification of recurrent GBM, with a surgical plan potentially reliant on the intra-operative pathological analysis of a tumor sample. Various canonical and emerging markers of GAMMs are explained below, as well as their evidence in rGBM (Table 1).

### 3.3. Microglia Markers

#### 3.3.1. P2ry12 and Tmem119

Histological staining of microglia in rGBM can elucidate the immune cell’s activation state, commonly discussed as a simplified model of “resting” or “active”. In the resting state, microglia are thought to be ramified, with very thin and long processes extending from their cell body. At this stage, increased expression of homeostatic core genes is of particular interest. Of the core microglia genes shown in Figure 1, P2ry12 and Tmem119 have gained particular interest due to the availability of commercial antibodies for histological staining. P2ry12 and Tmem119 are (1) selectively expressed in microglia compared to tissue-resident myeloid cells (e.g., macrophages) and (2) specifically located in a subpopulation of healthy microglia that are at rest [34,35]. Microglia downregulate their homeostatic genes, including P2ry12 and Tmem119, as they become activated [35] or under various pathological conditions, such as in many neurodegenerative diseases [36]. Unsurprisingly, the increased expression of P2ry12 in glioma is associated with increased survival, whereas a reduction in cytoplastic P2ry12 signal may correlate with a more severe glioma grade [37]. Furthermore, a TCGA-based analysis demonstrated that higher levels of P2ry12 expression correlated with a decreased risk of glioma recurrence [37].

These data suggest a potential clinical utility of both restoring microglia to a resting state in glioma and measuring P2ry12 and Tmem119 expression to assess the degree of reactive microglia inflammation. However, the degree of P2ry12 and Tmem119 expression on histology in rGBM remains less clear and has yet to be formally evaluated. Recurrent GBM likely has a highly activated environment in which most microglia remotely near the tumor have already begun to downregulate P2ry12 and Tmem119. Targeting the purinergic signaling cascade, a form of ATP-mediated extracellular signaling that facilitates microglia response to injury has gained interest in suppressing microgliosis [38] and may be relevant to rGBM as a means to prevent further reactive inflammation. However, further therapies should look to assess how to restore GAMMs back to a resting state alongside histological assessment of P2ry12 and Tmem119.

#### 3.3.2. Iba1

The ionized calcium-binding adaptor molecule 1 (Iba1, also “Aif1”) is the most traditionally and commonly used “microglia-specific marker”. Iba1 has been shown to label microglia across a variety of resting and activated states and largely differentiates Iba1+ cells from other glial cells of the CNS [39,40]. Compared to resting microglia, which stain diffusely positive for P2ry12 and Tmem119, activated microglia lose these markers as they become more ameboid in shape and demonstrate a less-ramified morphology, possibly facilitating functions like cell migration, proliferation, and phagocytosis at sites of injury [41]. In this context, Iba1+ staining labels both resting microglia and activated microglia as they downregulate their homeostatic genes. Thus, myeloid cells which are believed to be activated microglia are often labeled with double staining as Iba1+ and Tmem119- or P2ry12-, while double-positive cells (Iba1+/Tmem119+ or Iba1+/P2ry12+) indicate resting microglia [34]. Despite the literature commonly highlighting Iba1+ cells as microglia [42], it is important to clarify that Iba1+ may represent a more pan-myeloid-like cell marker, including both microglia and macrophages of various cell states [39]. This has caused many to label microglia and macrophages through double staining using Iba1 with other more specific markers of microglia and/or macrophages [34], which we expand on in the next section.

In recurrent glioma, the vast majority of cells are likely in an activated state following standard-of-care treatment and glioma re-growth [43]. A study of 91 paired primary and recurrent IDH-WT GBM samples demonstrated that Iba1+ microglia/macrophages were increased at disease recurrence [21]. This inflammatory transition was most apparent in primary tumors transitioning to a recurrent mesenchymal phenotype, possibly through macrophage-mediated processes like deactivating NF1, hypoxia, or by the mere number of activated microglia/macrophages in the TME. This mesenchymal and highly inflammatory phenotype is associated with a less favorable clinical outcome compared to other tumor/immune subtypes [21]. In another RNA-sequencing-based analysis of rGBM patients, increased Iba1+ expression correlated with increased transcription of various activated myeloid markers (e.g., CD68) and macrophage-like markers, such as CD206 and CD163 [44]. In line with these hypotheses, in rGBM mouse models, there is a significant increase in Iba1-positive cells. However, many of these cells do not co-express other microglia-specific markers, suggesting that the relative ratio of macrophages to microglia may be increased in rGBM as identified with Iba1, possibly competing for space within a hypoxic environment [45]. Using Iba1 labeling, these works highlight how the inflammatory landscape of rGBM is macrophage-rich as compared to what is seen in primary GBM.

An important drawback of using histological-based analyses of Iba1 alone is the likely context-dependent meanings of increased or decreased Iba1 expression. For instance, Kaffes et al. [46] classified primary and recurrent tumors into several tumor subtypes and found that high Iba1 levels conferred distinct prognostic implications contingent upon a pro-neural (worse prognosis with increased Iba1) or mesenchymal tumor (improved prognosis with increased Iba1) subtype (Table 1). Ultimately, Iba1 expression could have variable significance dependent upon tumor subtype [14,46], tumor location (e.g., tumor core versus infiltrated cortex at tumor margin) [18], or prior treatments [42]. Thus, broad Iba1+ staining and quantification in rGBM should be interpreted cautiously. Future work should, therefore, include Iba1 staining in rGBM tissue samples, specifically in combination with other markers of the TME, and careful consideration of the tissue biopsy location.

#### 3.3.3. Trem2

Originally a focus of research in neurodegenerative diseases like Alzheimer’s Disease with a potential beneficial role, triggering receptor expressed on myeloid cells 2 (Trem2) in glioma is believed to be widely expressed by disease-associated, inflammatory microglia. Although its exact significance is yet to be determined and remains controversial in glioma [47,48], its role in immunosuppression and phagocytic functions is under investigation. Trem2 in rGBM is an attractive marker given it may be highly expressed in an inflammatory glioma environment, possibly identifying microglia which has been driven to exhaustion. Although it is extensively implicated in microglia pathology, it is also co-expressed with various phagocytosis and macrophage markers (e.g., CD163 and Lyz) [47], suggesting macrophages or macrophage-like microglia are also histologically labeled by Trem2. Given these markers are histologically abundant in rGBM, Trem2 is an additional promising marker to examine if certain populations of microglia, or GAMMs in general, are differentiated by an inflammatory phenotype. Furthermore, given that Trem2-positive cells may identify a dysfunctional myeloid environment, staining for Trem2 on post-treatment tissue may allow for an assessment of the degree to which the rGBM immune microenvironment is responsive to therapy [49].

### 3.4. Macrophage, MDSC, and General Monocyte Markers

The few histological works that focus on GAMMs in rGBM suggest a shift from a predominantly microglia environment in primary GBM to one largely consisting of macrophages and/or macrophage-like microglia in rGBM. Based initially on other cancers in both clinical and preclinical models, the amount of infiltrating macrophages in the immune microenvironment is believed to be associated with a worse prognosis, and increased macrophage activity is associated with decreased overall survival [50]. In primary GBM and rGBM, this relationship has been less defined but is commonly assumed by the neuro-oncological community [18,51,52,53,54]. In patients with glioma, increased macrophage presence is associated with a higher grade of tumor and proliferative activity [52,53], increased vascularization [53], and decreased survival [18,51].

#### 3.4.1. CD68 and CD11b

CD68 and CD11b are common histological markers used to characterize inflammatory myeloid cells, including macrophages. CD68 is a glycosylated glycoprotein extensively expressed in tissue macrophages and monocytes [55,56]. However, activated microglia involved in phagocytosis can also express CD68 [57]. Thus, positive staining for CD68 alone generally indicates a highly inflammatory and active immune microenvironment rather than a specific cell population like macrophages or microglia [19]. CD68 has been extensively examined in glioma and recurrent GBM. In a novel trial of rGBM patients treated with a topoisomerase inhibitor, Topotecan, through convection-enhanced delivery, paired pre- and post-treatment analyses on tissue biopsies allowed for histological assessments of treatment response [19,58]. MRI-localized biopsies inside and outside of the drug infusion zone demonstrated significantly increased CD68+ myeloid cells after treatment. Bulk-RNA sequencing analyses on these biopsies demonstrated a significant correlation with the histological analysis, and post-treatment biopsies revealed increases in several pro-inflammatory cytokines and immunoreactive markers [19]. In other emerging treatments for rGBM, histological analyses of CD68 alongside RNA sequencing also revealed an inflammatory response following treatment [59].

Other studies have similarly demonstrated increased CD68+ cells in the rGBM microenvironment when comparing pre- and post-treatment glioma samples [3,18,60] and found it may be associated with decreased efficacy of immune checkpoint inhibitors, such as pembrolizumab anti-programmed cell death 1 (PD-1) [60] (Table 1). The authors report that anti-PD-1 therapy modulates macrophages in rGBM to a more pro-inflammatory phenotype, evidenced by some macrophages expressing MHC and others expressing markers of immunosuppression.

Interestingly, CD11b+, a myeloid cell marker with potential immunosuppressive properties (“myeloid-deprived suppressor cells”), is decreased after anti-PDL1 antibody treatment when the p38/MAPK pathway is inhibited in glioma-bearing mice with resistance to Temozolomide therapy—a common feature in rGBM patients [61]. Targeting this pathway leads to decreased CD11b+ myeloid cells due to decreased infiltration of microglia/macrophages. Thus, concurrent histological assessment of CD11b+ and CD68+ cells can facilitate an assessment of the degree of inflammatory myeloid cells that are infiltrating the brain parenchyma after treatment in rGBM. Immunofluorescence imaging of a checkpoint inhibitor-resistant murine glioma model showed a reduction in CD11b+ myeloid cells when treated with an antagonist of CCR2, a chemokine involved in inflammatory myeloid cell infiltration in the glioma microenvironment [62]. Other studies suggest that monitoring CD11b histologically can provide a biomarker of resistance in rGBM to immunotherapy agents [18]. Significantly increased CD11b+ cells were found in pre- versus post-treatment tissue in rGBM patients treated with antiangiogenic therapy, and the authors reported a CD11b+ subpopulation of macrophages, which may facilitate escape from antiangiogenic therapy [18]. After treatment, increased CD11b+ myeloid cells were found in both tumor bulk and infiltrative tissue and correlated with decreased survival in both regions.

Some have suggested that markers such as CD68 and other inflammatory myeloid markers cannot differentiate the primary versus recurrent GBM microenvironment [63,64] (Table 1). While appropriate characterization of GAMM histological staining likely requires context-dependent interpretation (e.g., according to spatial analysis in the TME and by tumor subtype), many studies consistently reported increased CD68 after various forms of therapy [18,19,60]. Therefore, these markers can aid in identifying the degree of post-treatment inflammation in rGBM.

#### 3.4.2. CD163, CD204, CD206

Canonical markers that are believed to be specific to macrophages and extensively expressed in rGBM include CD163, CD204, and CD206. These markers selectively label macrophages, which many believe to be polarized to an immunosuppressive state (“M2”) [65], and have been considerably studied in lung pathologies such as COPD [66]. In primary glioma, the ratio of these cells on histological analyses of paraffin-embedded glioma samples is associated with increasing histological grade and worse overall prognosis [52,67], as well as on RNA expression [68,69] (Table 1). Unsurprisingly, these markers show a high degree of correlation with CD68, but the proportion of CD163+ and CD204+ cells may represent only a subgroup of CD68+ cells, possibly reflecting a subgroup of macrophages or macrophage-like microglia in an immunosuppressive state [52].

CD163 is thought to be extensively involved in angiogenesis [70], a role that can facilitate tumor re-growth and invasion [71]. Increased CD163 staining on IHC was found to be an independent prognostic factor for patients with glioma and associated with increased risk of recurrence on both univariate and multivariate analyses [72]. In their study examining the effects of Temozolomide treatment with or without anti-PD-L1, Miyakazi et al. demonstrated that infiltration of CD163+ cells increased 2.7× in recurrent GBM specimens from patients treated with immunotherapy, as opposed to a very slight 1.1× increase observed in a pair of specimens from GBM patients treated with only standard therapy [73] (Table 1). In recurrent GBM patients treated with anti-angiogenic therapy, the level of CD163+ cells increased in pre-treatment versus post-treatment autopsy tissue. Unlike CD11b and CD68, CD163+ values did not correlate with overall survival [18]. Of note, the authors attempted to validate the specificity of IHC staining for these macrophages using flow cytometry with a variety of cell markers and reported a distinct cluster of GAMMs in the rGBM tumor, which were CD68+/CD11b+ in the tumor bulk and CD68+/CD163+ in the infiltrative margin of the tumor. The authors suggest a possible role of CD163+ in tumor cell and macrophage infiltration in rGBM [18].

CD204 and CD206 have not been extensively studied in rGBM but provide promising histological opportunities worth highlighting. CD206, a macrophage mannose receptor 1, has been considerably studied in tumor recurrence and metastases for other tumors [74,75]. Preclinical glioma studies using rGBM mouse models have demonstrated that CD206+ macrophages can remain for extended periods in the tumor microenvironment and sustain a protumorigenic phenotype (according to decreased co-labeling of MHC-II), which may facilitate tumor recurrence and aggressiveness [76,77]. Differently, CD204, also known as macrophage scavenger receptor 1 (MSR1), labels a specific group of macrophages found in areas of necrosis [69] and increased IL-6 signaling [78]. A recent study suggested CD204 was one of the only GAMM markers independently associated with worse survival and was also associated with an increased grade of malignancy, especially in IDH-WT glioma and mesenchymal GBM [69]. Histological assessments have been limited to primary GBM [78] and require further validation. Although not studied in rGBM patients in detail, histological assessment of CD204 and CD206 markers is likely important to examine the levels of post-treatment tissue response and inflammatory macrophage infiltrate.

#### 3.4.3. MARCO

A final marker worth highlighting is the macrophage receptor with collagenous structure (MARCO), which has a potential prognostic role and specificity for macrophage-like cells. In a large study conducted by our team on 66 glioma patients with single-cell RNA-sequencing of 98,015 cells and immunofluorescence analyses, 19,331 individual macrophages were profiled to examine how different macrophage subpopulations affect the glioma TME [51]. Through this unsupervised analysis, the scavenger receptor MARCO was associated with decreased overall survival and decreased time to recurrence. Of note, MARCO is a part of the same family as MSR1/CD206 (class A macrophage scavenger receptor, SR-A) and was similarly found to be associated with an unfavorable immune microenvironment, including mesenchymal traits and hypoxia, which can polarize macrophages to a pro-tumor phenotype [79]. Based on these data, MARCO may represent a particularly aggressive subpopulation of macrophages associated with poor prognosis (Figure 2). Importantly, when examining the effect of MARCO+ macrophages on immunotherapy responses and recurrence rates, a significant decrease in MARCO was found in responders pre- and post-PD1 checkpoint inhibitor therapy compared to non-responders [51]. Thus, histological assessments of MARCO present a promising opportunity to assess response in rGBM patients.

In this section, we discussed various canonical and emerging histological markers of GAMMs, and their relevance to and current uses in rGBM patients. It is important to note that GAMMs are plastic within a dynamically evolving environment. Cellular interactions can modulate how GAMMs may polarize into a pro-inflammatory “anti-tumor (M1)” or immunosuppressive “pro-tumor (M2)” state dichotomy. Significant research has been conducted on this topic, with various key markers identified that may suggest an M1 or M2 state [80,81,82,83] (Figure 1). Cytokine signaling and changes in RNA expression have been reviewed extensively elsewhere [84] and play a significant role in determining changes in GAMM polarization, as well as which and to what level histological markers are expressed. Histological-based assessments in this context can aid in characterizing the landscape of pro/anti-tumor phenotypes at any point in time. It is hard to use a single marker to truly capture the inflammatory landscape and potential degree of M1/M2 polarization; however, using many of these markers together with multiplex immunofluorescence or in parallel on adjacent tissue sections can provide a significantly increased amount of information.

**Table 1 cancers-16-03283-t001:** Studies characterizing changes in immune cells in recurrent glioblastoma patients using histological-based assessments. Histological-based assessments included immunohistochemistry (IHC) and/or immunofluorescence (IF) analyses. Result implications are drawn directly from each paper as the respective authors’ conclusions.

Study	Patient Population	Setting	Markers	Analysis	Study Result	Result Implication
Lu-Emerson et al., 2013 [18]	20 patients, rGBM	12 patients received antiangiogenic treatment and chemoradiation, 8 patients received chemotherapy and/or radiotherapy (no antiangiogenic treatment)	anti-CD68, anti-CD11b, anti-CD163, anti-CD14, anti-CD45, anti-CSF1R	IHC	Increased CD68 in tumor bulk (*p* < 0.01) and infiltrative regions (*p* = 0.02). Increased CD11b+ in tumor bulk (*p* < 0.01) and trend increase in I filtrate region (*p* = 0.09). Increased CD163 in tumor bulk (*p* = 0.09) in therapy group. Sequencing validated by IHC.	Inflammation induces mesenchymal-like state; macrophages are enriched in the vicinity of MES-like glioblastoma cells compared with OPC-like cells.
Gill et al., 2014 [2]	49 pGBM, 19rGBM	Characterized core vs. margins of p/rGBM	anti-CD44, anti-IBA1, anti-CD68, anti-SOX2	IHC	Core samples had higher cellularity and contained a higher amount of glomeruloid-type vascular necrosis than margins (*p* < 0.00001). Margin biopsies contained more NeuN+ neurons than core biopsies.	Non-neoplastic cells are a major component of the non-enhancing margins of p/rGBM tumors.
Wang et al., 2017 [21]	37 pGBM, 42 rGBM	Immune cell presence of p/rGBM	anti-AIF1, anti-NF1, anti-GeneTex	IHC	Increase in Iba1 (AIF1) at recurrence.	Increased inflammatory TME can drive mesenchymal-like tumor cell population.
Miyazaki et al., 2017 [85]	16 patients with both pGBM and rGBM samples	Molecular expression of immune environment in p/rGBM	anti-Ki-67, anti-TP53, anti-MHC class I, anti-MHC class II, anti-IDH-1R132H, anti-CD3, anti-CD8, anti-CD20, anti-CD45RO, anti-PD-L1, anti-Granzyme B, anti-PD-1, and anti-ATRX.	IHC	CD3, CD8, and PD-1 staining scores were significantly increased in rGBM specimens compared with pGBM specimens (*p* ≤ 0.05).	Stimulations, including AFTV treatment, induce the recruitment of many T cell type TILs, consisting mainly of CD8+ T cells. High CD8 and PD-1 scores after the secondary surgery were significantly poor prognostic factors of survival after second resection as high PD-1 score (*p* < 0.05 each), while high CD3 score trended as a poor prognostic factor (*p* = 0.065). In tumor markers, high PD-L1 grading trended as a favorable prognostic factor of PFS second resection (*p* = 0.095).
Rahman et al., 2018 [64]	38 pGBM, 12 rGBM	Immune markers of p/rGBM	anti-FOXP3, anti-CD70, anti-CTLA-4, anti-PD-L1, anti-PD1, anti-CD163, anti-CD68	IHC	No significant difference was identified in any immune marker between the primary and recurrent GBM.	No significant difference was identified in any immune marker between the primary and recurrent GBM.
Kaffes et al., 2019 [46]	48 pGBM, 8 rGBM	Mesenchymal vs. pro-neural GBM in pGBM and rGBM together	Anti-IBA1; anti-human FOXP3; anti-human CD8; anti-human CD3	IHC/IF	Mesenchymal subtype of GBM showed the highest presence of TAM, CD8+, CD3+, and FOXP3+ T cells.	High expression levels of FOXP3 and CD3G were associated with improved overall survival. High AIF1 expression levels confer a worse prognosis in the PN subtype but bestow a survival benefit in MES tumors.
Cloughesy et al., 2019 [86]	30 rGBM	15 rGBM patients treated with anti-PD-1 immunotherapy with SOC, 15 rGBM patients treated with SOC	anti-CD8, anti-CD45, anti-PD-1, anti-PD-L1	IF	Density of CD8+ T cells increased dramatically in the neoadjuvant (pembrolizumab) group.	Neoadjuvant administration of PD-1 blockade enhances the local and systemic anti-tumor immune response.
Liesche-Starnecker et al., 2020 [14]	21 patients with both pGBM and rGBM samples	Intra-tumoral heterogeneity and immune environment in p/rGBM	anti-EGFR, anti-GFAP, anti-Iba1, anti-Olig2, anti-p53, anti-Mib1	IHC/IF	Positive correlation of ALDH1A3 and Iba1 and increased levels of both in the progression (*p* = 0.000 and *p* = 0.001).	Temporal heterogeneity in GBM exists and potentially provides information important for prognosis and therapy resistance. For the recurrent tumors, a clear dominance of the mesenchymal/microglial-dominant subtype was observed.
Fu et al., 2020 [20]	13 pGBM, 3 rGBM	Quantification of TAMs in p/rGBM	anti-CD45, anti-CD68, anti-TNFa, anti-IDO	IF	CD4+/CD8+ T cells secrete more IL-10, IDO, TGFβ, T-bet, and TNFβ while expressing higher levels of PD-1, LAG-3, and TIM-3 than regulatory T cells, CD4+ T cells, and CD8+ T cells in p/rGBM	Provides further understanding of the immune environment of p/rGBM.
Miyazaki et al., 2020 [73]	6 paired p/rGBM	Standard radiochemotherapy and both standard and immunotherapy for assessment of preclinical models and pGBM and rGBM	anti-CD163	IHC	Infiltration of CD163-positive cells increased 2.7× in recurrent GBM specimens from patients treated with immunotherapy, although a 1.1× increase was observed in a pair of specimens from GBM patients treated with only standard therapy.	Anti-PD-L1 antibody treatment activates infiltration of CD163-positive Mϕ, usually considered as an M2 Mϕ marker, in a TMZ-resistant murine glioma model and also pGBM/rGBM tissue.
Tang et al., 2021 [44]	42 rGBM	LRRC15 characterization in the tumor environment	anti-LRRC15, anti-CD206	IHC	LRRC15 expression was positively correlated with CD206 expression in recurrent GBM (*p* = 0.001).	LRRC15 expression was positively correlated with CD206 expression in recurrent GBM. High expression levels of LLRC15 promote poor prognosis of recurrent GBM patients.
Magri et al. 2021 [63]	44 pGBM, 19 rGBM	pGBM (pre-treatment) and rGBM (post-treatment) with radiochemotherapy and temozolomide	anti-CD68, anti-CD8	anti-CD68, anti-CD8	In relapsing GBM, the presence of tumor macrophages redistributed in the two distinct areas, as the presence of BMDMs increased in the marginal area (*p* = 0.013) and MG decreased in the central zone (*p* = 0.002) Found no difference in CD68 cell density between primary and recurrent tumors; recurrent tumors demonstrated increased CD8+ cells (*p* = 0.015)	Increased recruitment of suppressive BMDMs in relapsing GBM.
Wang et al., 2022 [87]	13 pGBM, 11 rGBM (5 of each is the same patient)	Immune environment quantification in p/rGBM	anti-CD4, anti-CD8, anti-CD68, anti-PD-1, and anti-PD-L1	IHC/IF	Primary GBM typically had low levels of CD8+ T-cell abundance, and CD8+ T cells were sparse, isolated, and frequently confined to the perivascular space, while matched rGBM showed robust T-cell invasion of the cellular tumor.	We found that T-cell abundance was correlated with a significant increase in survival.
Alanio et al., 2022 [43]	14 pGBM, 13 rGBM	Immune environment in relation to survival in p/rGBM	anti-CD3, anti-Ki-67, anti-CD8, anti-Foxp3, anti-CD68, anti-CD163, anti-EGFR, anti-p53, anti-HLA-DR	IHC	Three groups: Myeloid I (tissue-resident microglial-derived TAM), Myeloid II (type-1 myeloid dendritic cells (cDC1), and CD163hi monocyte-derived TAM), Myeloid III (CD163low monocyte-derived TAM). Higher myeloid II and III in recurrent cases. Increase in the proportion of CD80-expressing myeloid cells detected in de novo tumors (*p* = 0.01) Similar overall content of CD8+ T cells in perivascular regions in paired primary and recurrent but increased number of perivascular regions associated with CD8+ T cells seen in rGBM samples. Perivascular T-cell high regions in rGBM displayed a lower density of FOXP3+ Treg cells and a significantly higher ratio of CD8+ T cells to FOXP3+.	Findings identify the spatial distribution of T cells rather than their abundance as a potential key immunological determinant that is associated with the evolution and pathogenesis of GBM. Enrichment of activated T cells in perivascular regions may be a determinant of longer survival in patients with rGBM T-cell compartment, especially in the perivascular regions of the tumor, in some patients with rGBM who may be polarized toward an antitumor-activated phenotype with clinical implications.
Al Dalahmah et al., 2023 [3]	45 primary and rGBM	Deconvolution of landscape in primary and recurrent GBM	anti-CD68	IHC	Specific cell types/transcriptional states colocalize in “tissue-states” defined by Sox-2, CD68, and NeuN staining. Significant enrichment in tissue state B (high CD68) gene signatures in rGBM samples compared to pGBM.	Compared to primary GBM, rGBM has increased CD68+ staining and a mesenchymal phenotype associated with worse survival.

## 4. The Inflammatory Landscape in the Context of Standard of Care Treatment

### 4.1. Immune TME Baseline Can Influence Glioma Resistance to Standard of Care Treatments

The standard of care (SOC) for GBM patients involves maximal surgical resection followed by chemotherapy and radiation therapy [1]. This includes the Stupp protocol, consisting of temozolomide (TMZ) and radiation therapy (RT). For the reasons mentioned above, the inflammatory landscape can influence resistance to treatment with TMZ and RT [88]. In a large meta-analysis of varying glioma datasets along with two independent cohorts, the level of Iba1 and CD68 after surgical resection and TMZ were found to be highly correlated with each other and also found to be correlated with survival status [72]. The authors report that lower levels of Iba1 and CD68 GAMMs were found in a matched patient cohort of long-term survivors (>2 years) compared to higher levels in short-term survivors (<1 year), suggesting that a highly inflammatory TME favored growth of a TMZ-resistant GBM. Subsequent in vitro studies suggest that GAMMs mediate the clinical response to TMZ in part through IL-11 secretion and activation of the JAK2/STAT3 pathway.

An immunosuppressive and TMZ-resistant tumor may act through the expression of CD74, an MHC II chaperone, and has been demonstrated in a number of studies [88,89,90]. Interaction between CD74 and macrophage migration inhibitory factor (MIF) leads to M2 polarization of intra-tumoral immune cells, with high MIF and CD74 on IHC correlating with resistant MGMT-methylated GBM tumors [89]. Therapeutics targeting inhibitors of MIF are being examined to suppress the inflammatory landscape and prevent resistant rGBM evolution [89].

Similarly, radioresistance in glioma is thought to be mediated through immune cells [91]. The mesenchymal phenotype of rGBM, a highly inflammatory subtype with increased Iba1 [21] and CD68 [3] histological staining, may increase resistance to radiation treatment through NF-kB and JAK-STAT signaling [92,93]. However, few studies have assessed how baseline immune markers on histology correlate with response to RT treatment to date.

### 4.2. TMZ Drives Inflammatory Changes in the Post-Treatment Setting

While the immune microenvironment of GBM can influence how a patient responds to TMZ and RT, these treatments can also induce a specific immune profile after treatment, which may contribute to the ability of rGBM to adapt to and resist further therapies [88]. TMZ is an alkylating chemotherapeutic agent with a robust ability to induce DNA damage in cells without high levels of base excision repair properties. Following some forms of chemotherapy in other cancers, M2 polarization of GAMMs promotes angiogenesis in the tumor, in part through VEGF secretion [80]. This can ultimately lead to tumor recurrence [94]. In GBM patients, the immune microenvironment response to TMZ varies [95]. However, those with an increased proportion of Iba1+ cells on histological assessments have significantly reduced progression-free survival and overall survival on regression analysis [95], possibly reflecting a less favorable mesenchymal shift compared to the low Iba1 patients, as mentioned in previous sections. Although the level of Iba1+ cells varied in this cohort, the degree of CXCL2-expressing GAMMs on histology significantly increased in tumors treated with TMZ compared to tumors not treated with TMZ [95]. CXCL2 is a known chemoattractant for GAMMs that is upregulated following TMZ [96] and is believed to promote tumor progression in glioma [97]. Inhibiting this pathway improves survival in GBM models [98] and provides an interesting therapeutic target.

Administration of chemotherapeutics and subsequent tumor cell death may alter the balance that neoplastic cells have established with other subtypes of immune cells and may promote the reduction or infiltration of these subsets [21,99,100,101]. Miyakazi et al. found increased numbers of PD-1/CD3/CD8 + cells in murine models that were then validated in human tissue in the recurrent setting [73]. In the literature, there are conflicting interpretations of the meaning of these fluctuations, but groups have claimed a correlation with poor prognosis for high levels of surface markers in the PD-1/PD-L1 axis (Table 1) [102].

### 4.3. Radiation Therapy Can Drive Glioma Toward a Resistant Phenotype

Radiation therapy utilizes high-energy wavelengths to induce DNA damage to proliferating neoplastic cells. Like chemotherapy, this treatment method can have unintentional effects on the TME and similarly induce a highly inflammatory immune microenvironment. This is thought to be due to the induction of hypoxia and the subsequent cascade of cytokine-induced recruitment of bone marrow-derived monocytes and other hematopoietic progenitor cells that are later able to develop into tumor-associated macrophages. In rGBM, for example, ionizing radiation has been shown to shift the GAMM composition toward macrophages rather than microglia [103]. Akkari et al. observed increased recruitment of TAMs and BMDM in rGBM after RT [103]. Interestingly, the authors report a similar proportion of CD68+ GAMMs in matched primary-recurrent patient samples but note a decreased level of P2ry12+ cells, suggesting a reduction in microglia or activation of GAMMs following RT. When comparing short- versus long-term relapsed tumors in primary GBM patients treated with RT, Wang et al. [21] found an increase in M2 macrophages and CD4+ T cells in short-term relapse GBMs, identifying the immunosuppressive homogeneity of post-treatment, pro-tumor macrophages and microglia.

### 4.4. The M2-Polarized GAMMs May Promote Recurrence in Post-Treatment Glioma

In the post-TMZ and RT setting, a milieu of inflammatory changes largely influences the behavior and phenotype of the residual tumor cells. An upregulation in “M2” immunosuppressive macrophages occurs in GBM samples after treatment with TMZ and/or radiation [104]. In preclinical models, TMZ treatment has been shown to increase surface expression of CD68 and CD206 markers [105], suggestive of what may be referred to as an M2-like myeloid phenotype, while irradiated tumors also demonstrate increased CD68+ cells in various tumor models [106]. This upregulation is negatively associated with prognosis and has been suggested to promote resistance to therapy in GBM patients [101]. Alanio et al. demonstrated T-cell enrichment in perivascular regions of rGBM compared to primary GBM [43]. Utilizing IHC, they also found increased activation of “macrophages” (measured with CD68) and decreased numbers of Treg in the rGBM setting [43]. Elsewhere, CD68 expression negatively correlated with T-cell activation [107]. As supported by various preclinical works, these data together suggest an expansion of CD68+ cells post-chemoradiotherapy facilitates an immunosuppressive TME, which can promote resistance to chemoradiotherapy. A deeper understanding of the onset, description, and actions of the post-treatment immune microenvironment of GBM is necessary to provide effective treatment.

## 5. Tissue Sampling of a Heterogeneous rGBM Tumor Microenvironment

To examine GBM in the recurrent setting, investigators rely heavily on the procurement of adequate human tissue specimens during surgical resection. These samples are fundamental to studying the rGBM inflammatory landscape because of the difficulty of recapitulating clinical rGBM features in vitro and in preclinical animal models. The inflammatory landscape can vary significantly throughout glioma-infiltrated tissue in both the primary [2] and recurrent settings [18]. For instance, in a post-mortem tissue-based assessment of rGBM patients treated with anti-angiogenic therapy, varying subpopulations of GAMMs preferentially accumulated in the tumor core and at the infiltrative margin, with only a subset of spatially defined markers correlating with patient prognosis [18]. However, methods of standard tissue sampling are highly inconsistent within and also between many neurosurgical centers. These challenges are further amplified in post-treatment recurrent glioma.

Firstly, tissue sampling is variable even if localized to targeted areas such as contrast-enhancing regions. The differing subtypes of gliomas within a tumor can lead to large heterogeneity in even spatially similar MRI-localized biopsies [21,30]. One study demonstrated that up to 62% of primary gliomas retain histologic features of multiple grades of tumors [16]. Patients with variable tissue sampling may even be excluded from clinical trials because of the difficulty of definitive identification and diagnosis of this challenging histology. This difficulty is amplified in downstream analyses. Histological analyses on biopsies obtained randomly in a heterogeneous GBM TME can vary significantly within a single patient and between studies comparing the landscape of various cells in the TME [12,16,108]. Liesche-Starneck et al. identified intra-tumoral heterogeneity within primary tumors as well as within paired-recurrent samples [14]. Highly mesenchymal biopsies of the rGBM TME demonstrate increased Iba1+ cells [14]; however, other parts of the tumor-infiltrated cortex with different tumor/immune profiles can have varying inflammatory infiltrate [5,18]. Extrapolating findings between studies can be difficult because no standard for biopsy location exists, such as from the contrast-enhancing core of the tumor or non-enhancing tumor margin, which also harbors heterogeneity. Studies have implemented methods to overcome some of the challenges in variability of tissue sampling. One solution is through radiographically guided tissue collection to accurately identify and sample different regions of the tumor landscape. In our experience, MRI-localized biopsies taken intraoperatively allow for an improved spatial resolution of tissue-based analyses through direct correlation of patient-registered images with pathohistological findings in both primary and recurrent GBM [2,3,19,87].

Another challenge is that in the recurrent setting, not all contrast enhancement is indicative of a tumor. Apart from tumor progression, the phenomenon of pseudoprogression can cause radiographic changes that resemble recurrence but consist of a significant amount of inflammatory myeloid cells alongside tissue necrosis rather than tumor cells [6]. Pseudoprogression has been found in up to 36% of surgical samples [109]. While histopathologic findings of pseudoprogression have not been standardized, studies reveal that samples of pseudoprogression are predominantly characterized by treatment-related changes, including radiation necrosis, chemotherapy-induced necrosis, fibrosis from surgical resection, microgliosis, and vascular hyalinization. The occurrence of pseudoprogression is not the only histologic challenge that arises when determining pathologic diagnosis. In the same cohort of patient samples demonstrating the rate of pseudoprogression, 4% of samples could not be definitively determined, likely associated with sample heterogeneity. This problem becomes particularly significant in selecting rGBM patients for clinical trials, as patients with pseudoprogression may have misleading responses to anti-tumor therapies. Given the difficulty in distinguishing pseudoprogression radiographically from true progression, improved histological analyses of immune cells can aid in patient selection.

Ultimately, human intraoperative tissue specimens provide an important avenue for the possible application of immunohistochemical and sequencing-driven prognostic analysis. The use of histological-based assessments of tissue biopsies, as displayed in Figure 2 with various markers, provides further insight into the TME changes after treatment and potential patient prognosis. The relative abundance of certain cell types or cell states can be quantified pre- and post-treatment and potentially provide prognostic information, such as increased macrophages post-treatment suggesting an inflamed and potentially poor prognosis [52,67]. Additionally, both bulk and single-cell RNA sequencing can provide additional information, nuance, and validation to these histological based assessments for quantification. In a large cohort, these quantifications together may be fed to a predictive analysis algorithm in order to provide prognostic value to individual tumors. Certain proteins have already been shown in the literature to have prognostic value. Chen et al. used MARCO bulk RNA sequencing to analyze expression across 603 rGBM patients. They found that when comparing high and low levels of MARCO expression, those with low levels of MARCO had increased overall and disease-free survival (OS: *p* = 0.0046, DFFS: = –0.018) [51]. Further describing the prognostic value of these markers, multiple studies have shown the promising prognostic value of identifying P2ry12 in tissue [37,110]. Future studies should focus on assessing tissue composition across multiple biopsies and incorporating additional analyses of the tumor microenvironment, such as RNA sequencing and radiomic profiling. Given the pronounced intra-tumoral heterogeneity that can exist even between adjacent biopsies, a more comprehensive mapping of the inflammatory landscape is essential.

## 6. Harnessing the Histopathologic Inflammatory Landscape for rGBM Treatments

The use of histological tissue-based assessments provides significant promise in neuro-oncology commensurate with the growing therapeutic interest in immunomodulatory agents. These assessments can evaluate the degree to which a targeted therapy stimulates or suppresses the desired specific cell population [62,89,111], modulates cell-cell interactions [112], or alters the immune microenvironment relative to treatment efficacy [18,60,73]. As previously discussed, immune cells provide constant cross-talk within the tumor microenvironment and can contribute to tumor progression and tumor resistance. Many therapies targeting glioma-associated microglia and macrophages are designed to overcome local immunosuppression. Ravn-Boess et al. found increased tumor killing in vitro and in vivo when using anti-CD97 antibodies to target the CD97 leukocyte adhesion marker, which has been implicated in immune activation [113]. Von Roemling et al. found that a combination of Temozolomide, anti-CD47, and anti-PDL1 conferred a survival benefit (in murine models) that was mediated by a robust innate and adaptive immune response [112]. Histological-based assessments in these studies, using IF and/or IHC, help confirm the appropriate immune targets of the drug were treated as well as immediate downstream effects.

There is also a growing interest in the immunomodulatory effects of standard treatments in glioma, including chemotherapy, radiation, and surgical intervention [114,115]. In a phase Ib trial of convection-enhanced delivery of topotecan, a robust CD68+ inflammatory response was found after treatment [19]. This may be a direct or indirect anti-tumor effect of the drug; however, tissue-based histological analyses provide an effective way to characterize the TME evolution after specific treatment and, therefore, identify additional potential therapeutic targets [59]. Several investigators have also explored the inflammatory response after treatment with TMZ [10]. By improving the understanding of both the antitumor effects of TMZ and its immune modulatory effects according to tissue analyses, improved combinatorial therapies that target immune cells can be devised, such as those that sensitize tissue to chemotherapy [116].

Along with therapies, recent studies have suggested demographic factors can influence the presence of a molecularly differentiated TME and, ultimately, patient outcomes. The incidence of GBM increases with age and eventually peaks between ages 75 and 84 [117]. However, gender too has been presented as a potential risk factor for the onset of GBM, with the male-to-female incidence ratio for diagnoses being 1.6:1 in some studies [118]. While the reasons behind this correlation remain unclear, many authors believe these epidemiologic differences may be attributed to the sex-dependent differences in major cell-type phenotypes in the TME, especially in GAMMs. Microglial sex differences may influence the degree of inflammatory response activation and resolution due to specific genetic drivers on the X chromosome. For instance, sex-specific roles of cell-adhesion molecules in the GBM TME, like *JAM-A*, have been correlated with preferential suppression of pathological microglia activation in females, whereas knockout led to larger tumors in female mice compared to males [119]. Differently, expression of the Xi-modulated gene *MPP1* may help regulate sex-biased inflammatory responses between male and female GAMMs. *MPP1* has been correlated with tumor grade in human microglia, and its immunosuppressive functions are thought to be preferentially demonstrated in male microglia and possibly lead to increased GBM growth in males [120]. Additionally, cell cycle inhibition proteins (CDKN1A, TP53, and RB1 pathways) are under downstream regulation of the X-chromosome with differential expression in various cell types between males and females, indicating a discrepancy between well-studied tumor suppressor pathways active in the brain [121]. Although the gene expression of sex-specific roles of different GAMMs remains to be further elucidated, they provide an important avenue to better understand molecular drivers of recurrence and potential targets in the TME.

Ultimately, high-quality pathological assessments and standard tissue-based assessments are crucial to better understand emerging therapies for rGBM. How the degree or lack of inflammation relates to treatment success is not clear, and these factors, described in Figure 3, are likely entirely context-dependent according to the treatment type, biopsy location relative to the tumor, demographic factors, and tumor subtype. Thus, it is imperative to obtain tissue after treatment, and even before when clinically available, to facilitate analysis of the cell-specific immune landscape in the context of rGBM treatment.

## 7. Conclusions

Technological advancements have revealed significant information about the cellular, genetic, and molecular characteristics of recurrent GBM. While these insights have spurred a number of emerging clinical therapies in rGBM, these strategies have yet to be optimized partly due to a poor understanding of the inflammatory immune microenvironment. In this review, we discussed the significant role of histological-based assessments of immune cells as a critical way to visualize and dissect the inflammatory landscape. Immunohistochemical and immunofluorescence analyses of varying immune cells, such as GAMM, can provide information on cell type and function within a shifting cellular milieu before and after treatment. However, these analyses have been extremely limited to date in rGBM. This work may serve as a guide on how to use various canonical and emerging histological markers to examine the rGBM immune landscape.

## Figures and Tables

**Figure 1 cancers-16-03283-f001:**
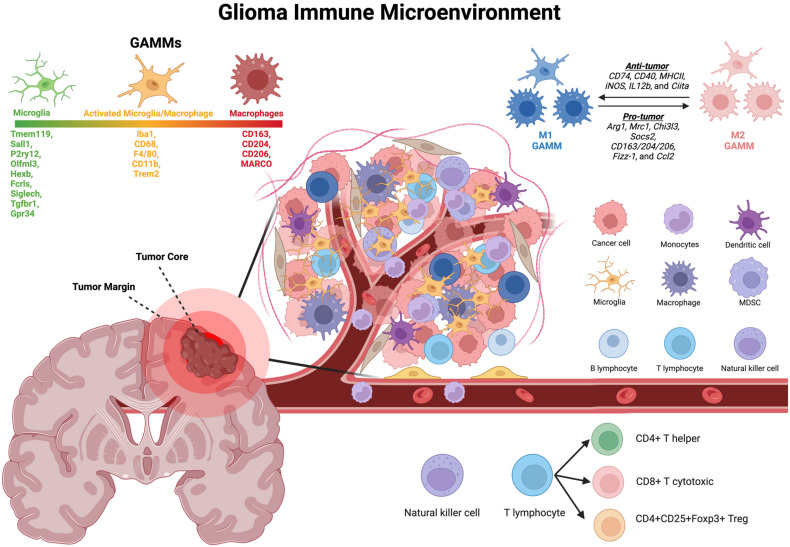
The recurrent glioma immune microenvironment. Immune cells influence the glioma microenvironment and dynamically interact and evolve. Changes in environmental signals, cell–cell communication, and spatial influences in reference to the tumor can all alter the inflammatory landscape in glioma patients. This figure was created through BioRender.com. Abbreviations: GAMMs = glioma-associated microglia and macrophages; MDSC = myeloid-derived suppressor cells.

**Figure 2 cancers-16-03283-f002:**
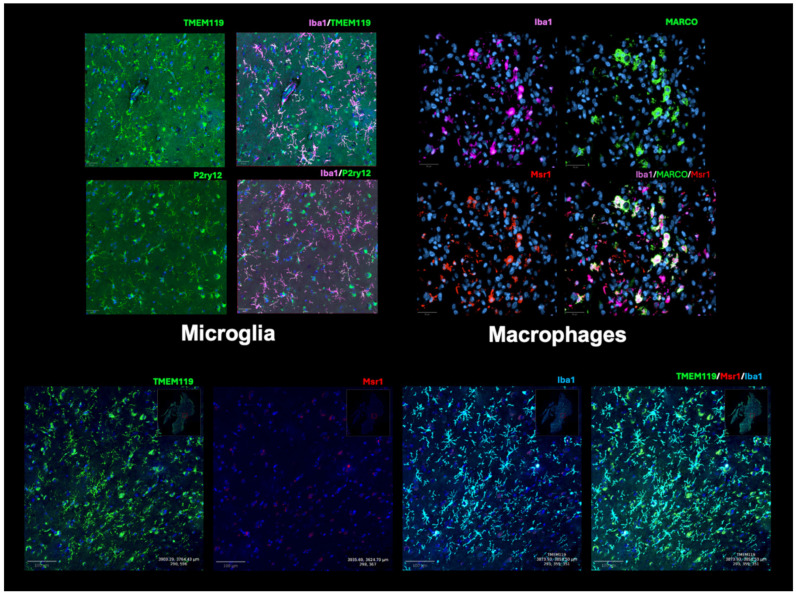
Common histological markers of GAMMs in rGBM samples according to immunofluorescence markers. In the top row, two examples are shown to label microglia-like cells (left) and macrophage-like cells (right). Iba1 is used in both examples as a pan myeloid marker, and then double-positive cells with Iba1+ and Tmem119+ or P2ry12+ are considered (resting) microglia compared to double positive Iba1+ with Msr1+ or MARCO+, which represent more macrophage-like cells. Note in the top right image, MARCO only labels a portion of Msr1+ cells, demonstrating a potential MARCO+ subpopulation of macrophages. The bottom row of the figures highlights the selectivity of these markers for different GAMMS, such that Iba1+/Tmem119+ cells do not overlap with Iba1+/Msr1+ cells. DAPI is used in all images in blue, while other markers are highlighted in the colors demonstrated on each figure panel. Markers used include P2ry12 (1:1000 anti-P2Y12 antibody, rabbit, Sigma-Aldrich, HPA014518, St. Louis, MO, USA), Tmem119 (1:500 anti-TMEM119 antibody, rabbit, Abcam, abcamab185333, Cambridge, UK), Iba1 (1:1000 anti-Iba1 antibody, chicken IgY, Aves Labs, 1BA1-0200, Davis, CA, USA), Msr1 (1:1000 anti-MSR1 antibody, mouse, Thermo Fisher J5HTR3, Waltham, MA, USA), and MARCO (1:1000 anti-MARCO antibody, rabbit, Invitrogen, PA5-64134, Waltham, MA, USA).

**Figure 3 cancers-16-03283-f003:**
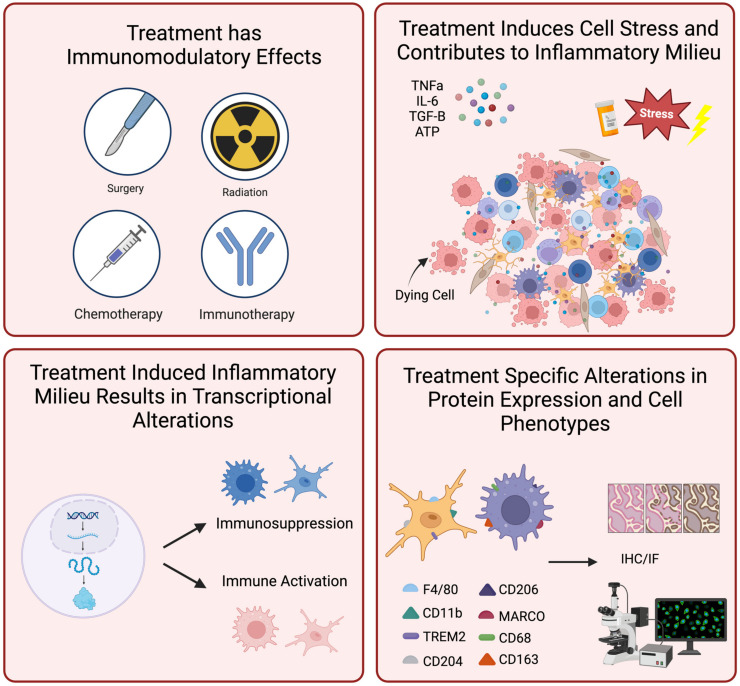
Standard-of-care therapies exert profound immunomodulatory effects on the TME. These treatments disrupt cellular components both directly and indirectly, creating a self-sustaining inflammatory milieu driven by complex signaling pathways, intercellular cross-talk, and cellular stress responses. This not only alters the peritumoral landscape but also drives intrinsic cellular changes by modulating transcriptional activity and gene expression. The effects of these changes can be effectively visualized and quantified through downstream immunohistochemical staining. This figure was created through BioRender.com.

## Data Availability

All data related to the current study are included in the current manuscript.

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
