# Peer review of "Unveiling the Inflammatory Landscape of Recurrent Glioblastoma through Histological-Based Assessments"

_cancers, 2024, doi:10.3390/cancers16193283_

Round 1

Reviewer 1 Report

Comments and Suggestions for Authors

This review analyzed the contribution of recurrent glioblastoma microenvironment to treatment response, shedding the light on the role glioma-associated microglia and macrophages. I found the review well organized and written but I have a minor concern, detailed below:

-since evidence supports gender and sex differences in GBM (see recent review Colopi A, Fuda S, Santi S, Onorato A, Cesarini V, Salvati M, Balistreri CR, Dolci S, Guida E. Impact of age and gender on glioblastoma onset, progression, and management; Carrano A, Juarez JJ, Incontri D, Ibarra A, Guerrero Cazares H. Sex-Specific Differences in Glioblastoma, etc ), it could be interesting to describe the differences in the tumor microenvironment and relative treatment response in these two classes

 - “Harnessing the histopathologic inflammatory landscape for rGBM treatments” a figure should be added

-the main body of the text should be reviewed since some typos have been found

-Authors should better explore  the possible application of newly described markers in the diagnostic process of glioblastoma, hypothesizing different potential prognostic algorithms

Comments on the Quality of English Language

None

Author Response

Please see attachment. Thank you for your review.

Reviewer 2 Report

Comments and Suggestions for Authors

Dadario et al. provide an extensive overview on the inflammatory landscape of recurrent GBM as assessed by histological analysis. This is an interesting topic and the manuscript is prepared very well.

To this reviewer, there is only one point that should be addressed in more detail. The topic of tumor heterogeneity and tumor-intrinsic properties that shape the TME is neglected a little bit and could be strengthened in the manuscript. The authors state that previous studies have examined paired primary-recurrent GBM samples to better understand the mechanisms of rGBM evolution and that rGBM also contains various tumor subtypes which are often retained upon recurrence. They cite the work by Hoogstrate et al (PMID: 36898379), that also suggests recurrent tumors show preferential mesenchymal progression. Since the mesenchymal subtype of GBM was described to display the highest percentage of microglia, macrophage, and lymphocyte infiltration vs other molecular subtypes elsewhere, it would be nice to add a brief discussion on how different molecular subtypes and genetic drivers in the (original vs recurrent) tumors could affect the immune landscape in the TME.

Author Response

(The authors gave the same response as above.)
